# Pluripotent Stem Cell-Derived Hepatocyte-like Cells: Induction Methods and Applications

**DOI:** 10.3390/ijms241411592

**Published:** 2023-07-18

**Authors:** Qiulin Luo, Nan Wang, Hanyun Que, Erziya Mai, Yanting Hu, Rui Tan, Jian Gu, Puyang Gong

**Affiliations:** 1College of Pharmacy, Southwest Minzu University, Chengdu 610225, China; qiulin1104@163.com (Q.L.); 17711380471@163.com (N.W.); quehanyun19960810@163.com (H.Q.); m18224067141@163.com (E.M.); h18185425548@163.com (Y.H.); 2College of Life Science and Engineering, Southwest Jiaotong University, Chengdu 610032, China; tanrui@home.swjtu.edu.cn

**Keywords:** induced pluripotent stem cells, hepatocyte-like cells, liver disease, induction methods, applications

## Abstract

The development of regenerative medicine provides new options for the treatment of end-stage liver diseases. Stem cells, such as bone marrow mesenchymal stem cells, embryonic stem cells, and induced pluripotent stem cells (iPSCs), are effective tools for tissue repair in regenerative medicine. iPSCs are an appropriate source of hepatocytes for the treatment of liver disease due to their unlimited multiplication capacity, their coverage of the entire range of genetics required to simulate human disease, and their evasion of ethical implications. iPSCs have the ability to gradually produce hepatocyte-like cells (HLCs) with homologous phenotypes and physiological functions. However, how to induce iPSCs to differentiate into HLCs efficiently and accurately is still a hot topic. This review describes the existing approaches for inducing the differentiation of iPSCs into HLCs, as well as some challenges faced, and summarizes various parameters for determining the quality and functionality of HLCs. Furthermore, the application of iPSCs for in vitro hepatoprotective drug screening and modeling of liver disease is discussed. In conclusion, iPSCs will be a dependable source of cells for stem-cell therapy to treat end-stage liver disease and are anticipated to facilitate individualized treatment for liver disease in the future.

## 1. Introduction

Hepatopathy is a major cause of illness and death around the world, and it carries heavy medical and economic burdens [1]. In China alone, approximately 300 million people are affected by acute or chronic liver diseases, such as viral hepatitis, nonalcoholic fatty liver disease, alcoholic hepatitis, and liver cancer [2]. Liver injury is a common pathological feature of all liver diseases. As liver damage continues, hepatic regenerative capacity no longer supports the repair of damaged liver tissues, and chronic liver disease eventually progresses to end-stage liver disease [3]. Liver transplantation is now the most effective treatment for end-stage liver disease [4]. However, less than 10% of the world’s need for liver transplants is now being satisfied due to a lack of liver donors [5]. Additionally, liver transplantation usually results in immune rejection [6]. Fortunately, regenerative medicine technology provides hope for discovering an alternative to liver transplantation as a treatment option.

Stem cells are promising tools of regenerative medicine to effectively treat a wide range of illnesses [7]. Multiple types of stem cells have been widely used in clinical practice, such as hematopoietic stem cells (HSCs), embryonic stem cells (ESCs), mesenchymal stem cells (MSCs), and induced pluripotent stem cells (iPSCs) [8,9,10]. Among them, iPSCs have contributed significantly to the field of regenerative medicine due to their distinct advantages [11]. iPSCs can circumvent genetically specific and immunological rejection since they can be produced from a variety of somatic cells, including those from patients with uncommon diseases [12]. Additionally, iPSCs can differentiate multidirectionally into three germ layers of cells [9]. iPSCs have already been widely applied in studies and treatments of inheritable arrhythmias, Alzheimer’s disease, liver diseases, and spinal cord injuries [11]. In the study of liver diseases, iPSCs have not only successfully created bioengineered human livers for transplantation but have also been used to create liver disease models to elucidate the related pathological mechanisms and to screen drugs [13,14]. In particular, how to efficiently generate hepatocyte-like cells (HLCs) with hepatocyte functions and properties from iPSCs in vitro has become a hot research topic.

In the present review, we compare the development of the embryonic liver with the in vitro induction of human iPSCs into HLCs, and the importance of simulating the microenvironment during liver development is emphasized. Additionally, the conditions required for iPSC differentiation into hepatocytes and related phenotypes to be monitored are reviewed and summarized. Furthermore, recent developments in liver disease modeling and drug screening using iPSCs are discussed. The present review may provide a methodological reference for the efficient differentiation of iPSCs into hepatocytes and promote related medical applications.

## 2. Advantages of iPSCs in the Treatment of Liver Disease

There are still many limitations to liver transplantation as the only effective treatment for end-stage liver disease; thus, there is an urgent need to discover new treatment options [15,16,17]. In vitro cultivation of primary human hepatocytes (PHHs) has emerged as a research area for the treatment of liver diseases. However, fresh or frozen PHHs do not perform well in the treatment of liver disease, because they die and lose liver function rapidly during in vitro culture, and are not readily available [18,19,20]. In 1998, human blastocysts were used by James Thomson’s research team to isolate ESCs for the first time. Because obtaining ESCs necessitates removing the inner cell mass (ICM) from the embryo sac, the origin and safety of human ESCs have been questioned [9,21]. iPSCs were created at the beginning of the 21st century; in 2007, Yamanaka’s group and Thomson’s group almost simultaneously reported the creation of the first human induced pluripotent stem-cell line [22,23]. Since researchers reprogrammed somatic cells to create iPSCs that can differentiate in multiple directions, research on iPSCs has gradually increased [22]. Patient somatic cells can be used to create iPSCs by inducing specific transcription factors, including c-Myc, Klf4, Oct3/4, and Sox2 [24]. iPSCs not only avoid ethical issues but also reduce the immune rejection that occurs during cell transplantation [25].

Additionally, animal models struggle to replicate the complexity of human diseases, while patient-derived iPSCs are genetically diverse, offering an opportunity to overcome related limitations [20]. Diseases caused by genetic diversity may be treated with individualized therapies based on patient-derived iPSCs. With the continuous advancement of biotechnology, the culture, induction, and differentiation of iPSCs in vitro have all gradually improved. An increasing number of induction methods for generating HLCs from iPSCs are proposed, which lead to the production of HLCs with liver functions and phenotypes comparable to those of mature hepatocytes. Additionally, iPSCs offer a great deal of potential for disease modeling and in vitro drug testing because they theoretically have the ability to generate a variety of cell types in the human body.

## 3. Development of the Liver in the Embryo

Humans develop from a single totipotent cell, the zygote, which produces a blastocyst with an inner and outer cell mass during development [26]. Among them, the ICM is the cell that embodies its pluripotency and gives rise to all the tissues of the embryo [27]. The ICM is arranged by cell movement to form three germ layers with multidirectional differentiation potential, namely, the endoderm, mesoderm, and ectoderm [28]. Progenitors within each germ layer specialize as development proceeds to give rise to particular tissues and organs. The dorsal–ventral and anterior–posterior locations of the progenitors in the embryo affect their identities [29]. As shown in Figure 1A, around week 4 of human embryonic development, the endoderm forms and divides into the foregut, midgut, and hindgut [20,29]. As embryonic development proceeds, the anterior foregut (AF) forms lungs, while the posterior foregut (PF) primarily forms the liver and pancreas, the midgut (M) forms the small intestine, and the hindgut (H) forms the large intestine [20].

As an important hub for numerous physiological processes, the liver performs a variety of functions such as glycogen metabolism and storage, regulation of bile transport, detoxification, and albumin secretion [30]. As mentioned earlier, the liver develops from the PF formed by the endoderm. By exploring the mechanisms controlling liver development in embryos, some insights can be provided into the induction from iPSCs to HLCs in vitro, which is critical for more efficient generation of functionally mature HLCs (Figure 1B). During liver development, the posterior foregut generates hepatic progenitor cells with bidirectional differentiation potential, which produce hepatocytes and bile duct cells under the influence of signaling factors secreted by the mesoderm, such as WNT, bone morphogenetic protein (BMP), hepatocyte growth factor (HGF), and fibroblast growth factor (FGF) [29,31]. The foregut secretes WNT antagonists during the development of hepatocytes, whereas the mesoderm around the midgut and hindgut expresses higher levels of WNT and FGF. Early foregut patterning depends on this gradient with low Wnt and FGF signaling [32]. Then, the foregut thickens to form a hepatic diverticulum, which is facilitated by the subsequent secretion of FGF from the septum transversum and BMP from the cardiac mesoderm [33]. The hepatic diverticulum’s basal lamina, which is abundant in adhesion proteins, ruptures at approximately 4–5 weeks of embryonic development, and then epithelial cells migrate to the septum transversum and differentiate into hepatocytes under the influence of the septum transversum mesenchyme [32,33,34].

## 4. The HLCs Induced from iPSCs

By simulating liver development in vitro, researchers have induced the generation of HLCs using iPSCs. How to produce HLCs with a primary hepatocyte phenotype and function is the key to the induction process, and various induction protocols are being developed with this aim [35,36]. Similar to ESCs, the current protocol for the induction of iPSC differentiation into HLCs involves three stages: the generation of a definitive liver endoderm, the production of hepatic progenitors, and finally, differentiation into reliable and functional HLCs [37]. Cell differentiation can be promoted by adding different cytokines or growth factors to stimulate relevant pathways at each stage of the induction protocol to mimic key signals during liver development in vivo. The cytokines or small-molecule compounds commonly involved in current experiments are summarized in Figure 2 [38,39,40,41,42,43].

### 4.1. Role of Cytokines/Small Molecules in Inducing iPSC Differentiation into HLCs

The early stage. Although the induction protocols of the different experimental groups may vary in detail, they all generally go through three phases. Activin A and WNT3 signaling is used to promote definitive endoderm (DE) formation in the early stages of iPSC-induced differentiation into HLCs, and Activin/Nodal signaling can control the expression of the iPSC-specific gene NANOG and, thus, contribute to the maintenance of iPSC pluripotency [44,45]. WNT signaling interacts with β-catenin and the endodermal marker SOX17 to promote FOXA2 expression. Activation of the WNT pathway can also inhibit PI3K, which is necessary to stimulate activin A for iPSCs to undergo differentiation [46]. The majority of induction regimens use activin A, a member of the transforming growth factor (TGF) protein family, to induce iPSC differentiation to DE in vitro by simulating Nodal control of Smad2/3 transcription. Nodal plays an essential role in DE production in vivo but it is extremely difficult to obtain highly active Nodal protein in vitro. Therefore, activin A is often used instead of Nodal to function in vitro. Moreover, high levels of activin A, typically 100 ng/mL, are used in the differentiation of DE because these levels might encourage the differentiation of iPSCs to DE, whereas low levels would preserve the iPSCs’ capacity for multidirectional differentiation [47,48].

Other research groups have found that small-molecule compounds can also enhance the differentiation of iPSCs toward the endoderm. For instance, the addition of LY294002, a PI3K kinase inhibitor, inhibits phosphatidylinositol 3-kinase signaling [39], or CHIR99021, a small molecule compound that activates the WNT/-catenin pathway [49,50], promotes DE differentiation together with activin A. In numerous recent studies, it was discovered that induction at high doses of CHIR99021 significantly promoted iPSC differentiation by activating the WNT pathway, whereas induction at low doses considerably increased the iPSC self-renewal ability [51,52,53,54,55]. It is obvious that the actions of the small molecules now under development are intended to stimulate WNT pathway activation or inhibit PI3K to encourage iPSC differentiation in concert with activin A.

The intermediate stage. DE is induced during the liver specialization phase using BMP and FGF. These cytokines are released by the nearby cardiac mesoderm and septum transversum mesenchyme during the development of the liver, and promote liver differentiation by activating GATA4 and repressing genes involved in pancreatic development [46]. Consistent with embryonic liver development, cell growth factors such as BMP and FGF have been used to induce hepatic progenitor cell production in vitro, and FGF-2, FGF-4, and BMP-4 are most frequently used [38,39,40,41,42]. Other studies have also stimulated liver differentiation during endoderm specification using dimethyl sulfoxide (DMSO) [56,57], HGF (hepatocyte growth factor) [56], or IWR-1, an inhibitor of the WNT/β-catenin protein pathway [58].

The final stage. Under various stimuli, liver progenitor cells have a bidirectional differentiation potential, producing bile duct cells or hepatocytes. HGF, oncostatin M (OSM), and the glucocorticoid dexamethasone (DEX) are necessary for differentiation. During the development of the liver, septal mesenchymal and endothelial cells release HGF, which binds to the tyrosine kinase receptor c-Met to improve cell survival and encourages hepatic progenitor cell proliferation [59]. OSM is an interleukin 6 (IL-6) family cytokine generated by hematopoietic stem cells that transmits molecules via phosphorylation signaling channels and is additionally able to transcribe activation factors during the maturation of hepatocytes [46,60]. At the same time, the combination of HGF and OSM factors with DEX increases the ability of hepatocytes to differentiate due to the fact that DEX upregulates the expression of mature liver-specific genes, such as albumin [38,39,41,42]. Furthermore, it has been revealed that certain small-molecule compounds play a beneficial role during the development of hepatocytes, such as SB431542, which blocks the TGF-β pathway [61], and compound E, which inhibits the NOTCH route [58], preventing the differentiation of hepatic progenitor cells into bile duct cells. Small-molecule compounds provide advantages over cell growth factors in terms of source stability because they are not easily subject to batch-to-batch variation.

### 4.2. Role of Other Additives in Inducing iPSC Differentiation into HLCs

In other aspects of promoting hepatocyte production, nutrients may aid in the induction of hepatocyte maturation in addition to mimicking the liver milieu to encourage induced differentiation of hepatocytes. Extracellular amino acids (AAs), for instance, play a significant role in the metabolic maturation of hepatocytes. Adding AAs in excess of what is necessary for nutrition actively promotes hepatic metabolism and functional maturation in vitro and facilitates the expression of cytochrome P450 biotransformation enzymes. In controlling the hepatic transcriptional network, AAs perform a similar role to growth hormones in driving liver differentiation [62]. To improve the functional maturation of HLCs, especially the expression level of drug-metabolizing enzymes, Takayama and colleagues [63] established the CYP3A4–NeoR–EGFP transgenic reporter human iPS cell line (CYP3A4–NeoR–EGFP iPS cells) using genetic programming technology. They then treated HLCs differentiated from CYP3A4–NeoR–EGFP iPS cells with neomycin and discovered that the positive expression of CYP3A4 increased to more than 80%, compared with only approximately 20% in previous studies. Interestingly, they discovered that neomycin treatment greatly boosted the expression of other mature liver-specific genes, as well as a number of drug-metabolizing enzymes in HLCs [63]. In addition to upregulating the expression of liver-specific genes, enhancing differentiation efficiency at each stage of the hepatocyte differentiation process is also important for the production of hepatocytes. In this regard, it was discovered that doxycycline increased the efficiency of endoderm differentiation by preventing apoptosis and did so via the protein kinase B pathway during the differentiation of iPSCs into endoderm, but that there was no appreciable difference in gene expression and phenotype relative to hepatocytes generated without the induction of doxycycline [64,65].

### 4.3. The Three-Dimensional (3D) Culture System Recreates the Liver Microenvironment

The majority of earlier studies on the induction of iPSCs into hepatocytes involved iPSCs differentiating into final endodermal cells and then continuing induction into monolayer hepatocytes. However, because the process of liver development is controlled by the cooperative participation of many cells and tissues, these techniques have some limitations [66,67,68,69]. In recent years, experimental protocols for the in vitro induction of iPSC differentiation into HLCs have been repeatedly validated, and many new approaches have been proposed to improve the maturation and functionality of HLCs. The key to achieving this goal lies in more faithful replication of the liver microenvironment. Newly developed culture strategies mainly include using specific extracellular matrices or coculturing with other liver parenchymal cells.

#### 4.3.1. Induction of Differentiation Using Specific Extracellular Matrices or Culture Platforms

The 3D culture system can produce a protein-based 3D scaffold that resembles the microenvironment of the human liver, allowing different types of liver cells to coexist and differentiate together and then receive various signaling molecules to eventually develop into hepatocytes in a coordinated manner [70]. In 3D culture systems, HLCs have been produced using extracellular matrices containing collagen, fibronectin, or vitronectin [20]. Pluripotent stem cells have been differentiated using Matrigel in numerous studies, but a recent study found that laminin 521/111-differentiated HLCs displayed higher induction and P450 activity than those of their control counterparts [71]. Laminin is a favorable option for hepatobiliary cell codifferentiation induction protocols because of the advantage of improved support for biliary tubular structures demonstrated by it in the study’s results. Additionally, iPSCs were cocultured in spheroids [72] or microenvironments [73], which led to the production of more functionally mature hepatocytes than the usual two-dimensional (2D) culture environment, indicating that the 3D culture system can better mimic hepatocyte function.

In contrast to conventional 2D culture techniques, Gieseck [74] and his associates found that inducing the formation of HLCs in 3D collagen matrices dramatically improved the function of HLCs. When cells were differentiated in this culture method, polarized structures were created, which increased the drug metabolism rate of HLCs. Additionally, this polarized structure lengthened the time that HLCs maintained their function in vitro to 75 days. This may be because polarized cells have a higher rate of drug metabolism, which makes them correspondingly more capable of handling toxins. Recently, a study [56] created a new differentiation protocol for stem cells based on 3D culture that also produced cells with polarized structures. The researchers induced the generation of columnar polarized HLCs on transwell filters. The transwell filters’ permeability enables basal and apical cellular uptake and the secretion of molecules, and allows metabolic activity to more closely resemble physiological activity. The HLCs produced by the current experimental protocol lack critical in vivo features, such as cell polarity. The 3D culture-based study described above overcomes this deficiency by producing polarized HLCs that can secrete cargo in a directed manner, with the basal layer secreting albumin, lipoprotein, and urea, and the top layer secreting bile acids [56]. This powerful cell culture system not only increases the rate at which HLCs metabolize drugs but also offers a reference value for the development of new drugs for the treatment of liver illnesses.

Additionally, the 3D-HLC spheroids produced from the microporous culture platform also showed greater CYP3A4 enzyme activity and maintained high levels of hepatocyte nuclear factor (HNF4) expression. The sensitivity of 3D-HLC spheroids to toxic compounds in the liver was also improved in comparison to the 2D culture technique due to the higher numbers of drug-metabolizing enzymes [75]. Therefore, 3D culture has been used to generate tissue organs differentiated from iPSCs on a large scale for in vitro drug toxicity assessment [76].

#### 4.3.2. Coculture with Other Cells

Although the aforementioned culture method demonstrated the benefits of the 3D system in the culture of HLCs, further study is required to produce fully functional mature hepatocytes. For instance, a number of experiments have been conducted to determine if coculturing with other liver cells enhances the development of hepatocytes [41,77,78,79]. The liver contains not only hepatic parenchymal cells that perform major metabolic and synthetic functions, such as hepatocytes and bile duct cells, which are in direct contact with the blood through the hepatic sinusoids, but also many nonparenchymal cells that are essential for maintaining organ function and structure, such as Kupffer cells, hepatic stellate cells, and liver sinusoidal endothelial cells (Table 1) [30]. Hepatic parenchymal cells make up approximately 70–80% of the liver mass. The membrane polarity of hepatocytes and their interaction with nonparenchymal cells allow for proper liver function. Hepatocytes are tightly packed together to produce bile ducts, which are responsible for transporting bile [80]. The liver sinusoidal endothelial cells are connected to the basolateral surface of hepatocytes, allowing communication between the liver parenchyma and the blood. Kupffer cells—the resident macrophages of the liver—reside in the sinusoids, while hepatic stellate cells reside in the space of Disse between the hepatocytes and the sinusoids (Figure 3) [34].

In a recent study, iPSCs from a variety of healthy and steatohepatitis patients were used. They were cultured with retinoic acid (RA) for 4 days after being induced to form a well-defined endoderm. Next, hepatocytes were induced using hepatocyte maturation medium, and the result was the development of multicellular hepatocyte-like organs (HLOs) [77]. Using immunofluorescence staining and single-cell RNA sequencing, the researchers were able to identify the various cell classes present in liver-like organs. Their findings revealed that 59.7% of the cells expressed the hepatocyte markers HNF4A, CEBPA, and RBP4, while 31% of the cells were enriched for the stellate cell markers VIM, CYGB, and DES, and 5.1% of the cells expressed bile duct cell markers. Furthermore, they found positive expression of Kupffer cell markers, and these findings imply that they co-differentiated a variety of hepatocyte types to stimulate the development of liver-like organs. A fold increase in CYP3A4 activity in HLOs in the rifampicin-treated group and a fourfold increase in the percentage of vitamin A^+^ stellate cells in the retinol-treated group are two examples of results that coincide with hepatocyte-specific functions in vivo and show that the HLCs induced by the research team produced functional hepatocytes [77]. This codifferentiation of hepatocyte-, stellate-, and Kupffer-like lineages in a PSC-derived organoid culture is advantageous to research on the inflammatory pathophysiology connected to steatosis.

On the basis of the close association of multiple cells during liver development, Wu [41] and his team established a 3D culture system in which they induced the first functional hepatobiliary organ models from human iPSCs. These organs had hepatocyte functions and survived after being transplanted into immunocompromised mice for more than 8 weeks. This is most likely due to the codifferentiation of HLCs and cholangiocyte-like cells. Vallverdu [78] and his colleagues recently achieved the differentiation of iPSCs into hepatic stellate cells with functional characteristics and the phenotype of primary hepatocytes, most of which express PDGFRβ. Furthermore, they cocultured them with hepatocytes to create three-dimensionally structured hepatic spheroids. Their discoveries will provide a source of cells for in vitro liver fibrosis research and toxicity testing [78]. In summary, the possibility of cultivating many hepatocyte types in 3D medium is also an indelible boost for developing complex in vitro liver systems to simulate the liver microenvironment and treat liver illnesses. In a different study [79], it was shown that a polyethylene glycol hydrogel microenvironment supported the coculture of iPSC-induced hepatocyte production together with all liver nonparenchymal cells and maintained the function of each cell for at least 1 month in vitro. With regard to modeling the in vitro pathological models of hepatitis and liver fibrosis, the cells produced by this hepatocyte maturation gel coculture system fared better than did those produced by each cell type when cultivated separately [79]. This finding supports that using iPSC-induced hepatocyte generation alone to model liver disease is insufficient to capture the subtle changes that occur in the in vivo microenvironment after the onset of liver disease.

#### 4.3.3. 3D Bioprinting

A recent development in tissue engineering called 3D bioprinting makes it possible to precisely manufacture large parenchymal organ structures, which also provides new opportunities for liver regenerative medicine [90]. Researchers found that 3D printing human iPSC-derived parenchymal cells into spherical aggregates greatly enhanced liver function and prolonged tissue survival time in vitro by using an alginate/hyaluronic acid hydrogel combination [91]. In another study [92], the researchers created a microscale hexagonal architecture and implanted hiPSC-derived liver progenitor cells inside it together with human umbilical vein endothelial cells and adipose-derived stem cells. A distinct hepatic progenitor cell phenotype and improved liver function were observed after many weeks of in vitro culture. These improvements included higher levels of liver-specific gene expression and induction of cytochrome P450 [92]. These 3D printing techniques based on stem cell-derived cells are able to facilitate the in vitro maturation and functional maintenance of HLCs in a bionic microenvironment. Thus, the model has great potential for pathophysiological studies, disease modeling, and early drug screening.

In summary, it appears that the induction process is difficult regardless of whether it uses 3D culture, conventional cytokines, or other small molecules that encourage the formation of hepatocytes. It is important to consider how to quickly and readily produce a high number of hepatocyte liver cells with liver function for clinical studies because these induction factors are very expensive, placing a significant financial burden on preclinical studies.

## 5. Identification Methods of HLCs Differentiated from iPSCs

For the effective production of functionally mature HLCs, the identification of the morphology, phenotype, and liver function of differentiation-generated HLCs is crucial. Despite the fact that numerous studies have been performed on the production of HLCs from iPSCs, there is no set test standard. We summarize the common tests that are currently used to evaluate the quality of HLCs. And regardless of the test, PHHs are the gold standard for testing because they are morphologically, phenotypically, and functionally highly comparable to healthy human liver cells [19,93].

In terms of morphology, HLCs should resemble primary hepatocytes in that they should be polygonal in shape and have a rising nucleus-to-cytoplasm ratio as differentiation progresses and the nucleus grows larger [94]. Moreover, hepatocyte marker genes and proteins are frequently detected in HLCs using protein immunofluorescence, quantitative real-time PCR, and Western blot analysis [82]. Currently, iPSC-induced HLCs resemble a fetus, most likely at a stage between the end of the first trimester of fetal embryonic development and adult hepatocytes [95]. The functional characteristics of HLCs compared to adult hepatocytes were deficient, as evidenced by the expression of the fetal marker proteins AFP and CYP3A7 by HLCs, while CYP enzyme lines that are specifically expressed in adult hepatocytes, such as CYP3A4, were expressed at much lower levels in HLCs than in primary hepatocytes [96,97,98]. Additionally, HLCs had a lower expression of the AQP6 protein than that of primary hepatocytes, and they had a reduced capacity to secrete urea in comparison to controls. AQP6 is a membrane channel protein that facilitates urea, water, and glycerol transport and plays a crucial regulatory role in the urea cycle. A reduction in AQP6 expression causes urea to build up intracellularly. Notably, AQP6 expression was minimal in embryonic liver tissue [95]. Thus, it appears that HLCs are more similar to the developing embryonic liver in the expression of many genes.

To ascertain the differentiation effect of HLCs, the expression level of mRNA was primarily assessed by quantitative real-time PCR and compared with primary hepatocyte-specific genes. Among them, ALB, which is synthesized by mature hepatocytes, is frequently utilized as a reliable marker protein for liver function testing [14,39]. Several marker proteins, such as AAT, HNF4, CK8, CK18, AFP, and the CYP450 enzyme system, have also been employed as markers [14,38,62,99]. However, AFP is a fetal hepatic protein. The predominant drug-metabolizing enzyme expressed by HLCs is CYP3A7, while the expression of CYP3A4, a marker protein of primary adult hepatocytes, is low [96,97,98]. Lastly, the positive expression rate and specific expression location of the tagged proteins were confirmed by protein immunofluorescence staining and corroborated with the results of RT-qPCR. The detection of hepatic functional properties should also be taken into consideration to determine the maturity of HLCs. Hepatocytes have more than 500 different functions [100]; Table 1 lists the relatively significant liver-specific functions that have been frequently tested in recent studies.

## 6. Applications of HLCs Differentiated from iPSCs

iPSCs are used to study the pathological mechanisms in the development of diseases by virtue of their source diversity and gene specificity. Patient-specific disease models can be established, which not only provides a solution for personalized patient treatment but also holds promise for in vitro drug screening.

### 6.1. Modeling Liver Disease by Human iPSCs

The original organ or tissue is the best source of cells for in vitro disease modeling; nevertheless, primary cells from diseased tissues of patients are sometimes difficult to obtain and cannot maintain their original functional features when cultivated in vitro [101]. iPSCs have no source restrictions and can be obtained from patients with a variety of diseases, allowing them to model many liver diseases with the specificity of human primary cells. Additionally, the genetic background influences disease-associated cell phenotypes, and iPSCs can be used to determine the degree of the effect [102,103,104]. Hence, iPSCs have great potential in understanding complex genetic interactions in vivo under the influence of disease, including genetic interactions in patients with developmental disorders or congenital diseases [49,105,106]. Table 2 lists the hepatocytes produced by conventional 2D culture techniques and 3D-like organs for modeling liver disease.

In previous studies, it was demonstrated that hepatocytes produced from iPSCs created utilizing a 2D differentiation strategy are efficient in the treatment of several liver disorders [49,107,108]. With the creation of iPSC-derived organoids, disease modeling with iPSCs has made a huge stride forward [105,109,110]. Organoids are stem cell-derived 3D multicellular aggregates that self-organize during differentiation to mimic the structural characteristics and cell–cell interactions of mature tissues [111].

More recently, Ramli and colleagues [40] used a culture system with a specific chemical composition to induce the development of liver organs with hepatocytes and bile cells. The differentiation of hepatocytes made from iPSCs resulted in a continuous network of bile ducts, which was not observed in earlier studies. The bile duct network plays an important role in bile transport and detoxification, and the group developed the first in vitro model of drug-induced cholestasis using the antidiabetic drug troglitazone to cause acute disruption of bile acid flow, which provides an additional resource for studying in vitro modeling of liver disease caused by cholestasis [40]. The hallmark proteins albumin and cytokeratin were measured to assess cell differentiation into hepatocytes and biliary tract cells. The differentiation produced functional cells, as evidenced by the presence of hepatocytes with albumin-secreting and cytochrome P450 activity and biliary tract cells with γ-glutamyl transferase and alkaline phosphatase activity [40]. The researchers produced liver-damaged organs using a medium containing free fatty acids, and analyzed the gene expression between liver tissue from individuals with nonalcoholic steatohepatitis (NASH) and liver-damaged organs. They discovered that the profiles were similar which indicated that the produced liver organs may be able to replicate NASH in vitro.

Recently, liver diseases caused by urea cycle disorders (only ARG1 deletion, ASS deletion, and citrulline deletion) have been modeled with hiPSC-Heps, while in vitro modeling of ornithine transcarbamylase deficiency (OCTD), which is the most common cause of urea cycle disorders, has rarely been reported [112,113,114]. The reason for this is most likely due to the low urea production of iPSC-induced HLCs used in recent experimental studies. One study [95] restored the urea-secreting function of HLCs by forcing the expression of AQP6. An in vitro OTCD model was developed employing patient-derived iPSCs with the typical liver disease profile of OTCD patients after overcoming the defect of reduced urea secretion in earlier studies. This induced production of HLCs from patient-derived cells is beneficial for the establishment of targeted therapeutic regimens.

Furthermore, contrasting the differences in the pathological symptoms of liver disease models developed by patient- and healthy person-derived HLCs can help to personalize the disease treatment plan in the future. In this regard, a team of researchers developed iPSCs from liver-like tissues from Wolman disease patients, who show a significant build up of lipids in hepatocytes that can result in steatohepatitis or liver fibrosis in severe circumstances [77]. The analysis of the differences in lipid degeneration between non-Wolman-derived HLOs and Wolman-derived HLOs showed more severe lipid degeneration in Wolman-derived HLOs, which supports in vitro modeling and steatohepatitis therapy.

Building in vitro models of liver disease using human iPSC-induced HLCs has been successful, allowing researchers to better understand the disease’s pathophysiology and to develop new therapeutic treatment strategies [115]. Three-dimensional organoids are the evolution of two-dimensional culture that represent a more mature establishment of liver disease models, providing a reliable basis for the exploration of the mechanisms of various complex liver diseases and personalized treatment.

### 6.2. Hepatoprotective Drug Screening via Human iPSCs

In addition to disease modeling for the study of disease mechanisms, iPSC-based models can also be used for drug screening to discover new therapeutic approaches. iPSCs hold promise for in vitro drug testing by virtue of their source diversity and genetic specificity. Through the development of a liver organoid using iPSCs from Wolman patients, researchers have discovered possible therapeutic benefits of OCA and FGF19 in Wolman disease [77]. LAL activity is decreased in Wolman disease, and LAL deficiency results in increased ROS production, lipid accumulation, and excessive mTOR activation. In contrast, FGF19 inhibits lipid accumulation while also enhancing the survival of HLOs derived from Wolman patients, and ROS levels are significantly decreased [77]. These findings not only confirm the therapeutic effect of FGF19 on Wolman disease but also demonstrate that HLOs produced by iPSCs show genetic specificity for Wolman patients and may be more widely used for in vitro drug screening of other genetic illnesses in the future.

It is also important to note that adverse drug therapy reactions can result in major health issues that affect a number of bodily organs and cause disease or even death. One of these unfavorable effects is drug-induced liver injury (DILI), which accounts for more than 50% of cases of acute liver failure and is a common reason for the withdrawal of drugs from the market [116]. Medications that may be harmful in clinical trials can be predicted in vitro, and the possibility of liver toxicity in clinical trials will be significant if therapeutic dosages of liver disease drugs are cytotoxic in these tests [32,117]. According to Shinozawa and colleagues, 238 commercially available medicines were tested for their propensity to create drug-induced liver injury using an in vitro toxicity screening method based on iPSC-derived liver-like organs. By identifying drug activity, the potential to produce cholestasis, and/or mitochondrial toxicity, among other things, the liver organ-based toxicity test method they developed can be used to evaluate the likelihood that medications would damage the liver. The results have high predictive value [39]. Another instance of the validity of in vitro drug testing was that all substances triggering the creation of xenobiotic metabolizing enzymes in human livers were successfully detected by in vitro human hepatocytes [32].

HLCs produced using the current methodology nevertheless differ significantly from PHHs, posing substantial constraints for in vitro toxicity assessment. PHHs continue to be the gold standard for studies on liver disease [118]. The levels of drug-metabolizing enzymes have a significant impact on the outcomes of toxicity studies conducted on cells, especially the cytochrome P450 enzyme system [32,116]. Therefore, the hepatocyte system created via the induction of iPSCs still needs to be improved to achieve the upregulation of certain genes that are not fully expressed in HLCs or the downregulation of other genes that are not typical of hepatocytes. Examples include optimizing the microenvironment for cell growth and the way that cytokines are combined. Through the use of a 3D microporous culture platform, Lee was able to induce iPSCs to produce 3D-HLC spheroids that expressed higher CYP4A3 enzyme activity and increased sensitivity to toxicants in the liver compared to those of hepatocytes produced in 2D culture and may be an effective tool for in vitro drug toxicity testing [75].

In conclusion, obtaining HLCs with a functional maturity level that is more similar to that of primary hepatocytes is a continuing necessity for advancements in drug testing and in vitro screening. The potential of HLCs in drug testing is crucial to making it easier to explore drug treatment mechanisms, optimize compounds, and screen for drugs.

## 7. Conclusions

Research to reprogram human somatic cells into iPSCs and direct their differentiation into HLCs offers hope for treating patients with liver failure. To enhance the maturation and functionality of hepatocyte-like cells, numerous new induction protocols based on conventional 2D induction systems have been proposed, such as coculture with other cell types [82,119] and 3D culture using a gel matrix [79]. However, embryonic liver development is subject to multiple cellular interactions and co-development in vivo. The challenge of creating this cellular milieu for the in vitro for the generation of HLCs affects the morphology and function of the resultant HLCs [120]. Furthermore, due to the heterogeneity of the cellular product created from iPSCs for therapeutic application, there can be issues with tumorigenicity in vivo [121]. Despite a significant improvement in HLC function, the current technique still falls short when compared to PHHs and may require a more precise recapitulation of the complicated microenvironmental changes and real-time signal regulation in the developing hepatic microenvironment. After overcoming these difficulties, the genetic specificity and source diversity of iPSCs will offer highly promising methods for personalized treatment of liver disease.

## Figures and Tables

**Figure 1 ijms-24-11592-f001:**
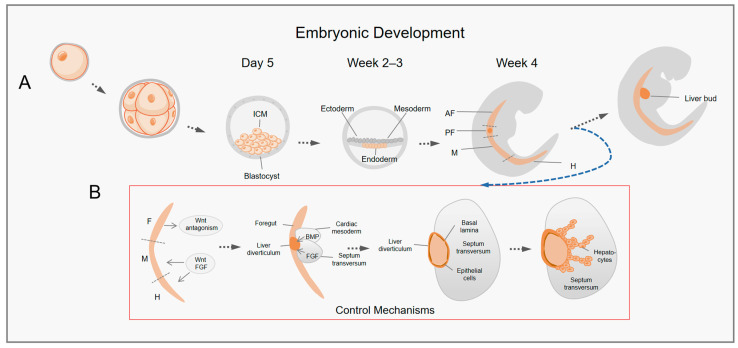
The key steps of embryonic development and the control mechanisms of liver development. (**A**) The developmental states presented at different time stages of the embryo. (**B**) The key control mechanisms of liver formation during embryonic development. ICM, the inner cell mass; AF, anterior foregut; PF, posterior foregut; F, foregut; M, midgut; H, hindgut; FGF, fibroblast growth factor; BMP, bone morphogenetic protein.

**Figure 2 ijms-24-11592-f002:**
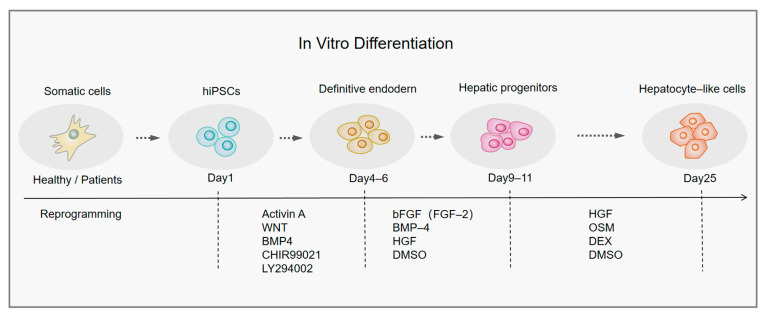
The key steps of in vitro differentiation protocols from iPSCs to HLCs. hiPSCs, human induced pluripotent stem cells; BMP4, bone morphogenetic protein 4; FGF, fibroblast growth factor; HGF, hepatocyte growth factor; DMSO, dimethyl sulfoxide; OSM, oncostatin M; DEX, dexamethasone.

**Figure 3 ijms-24-11592-f003:**
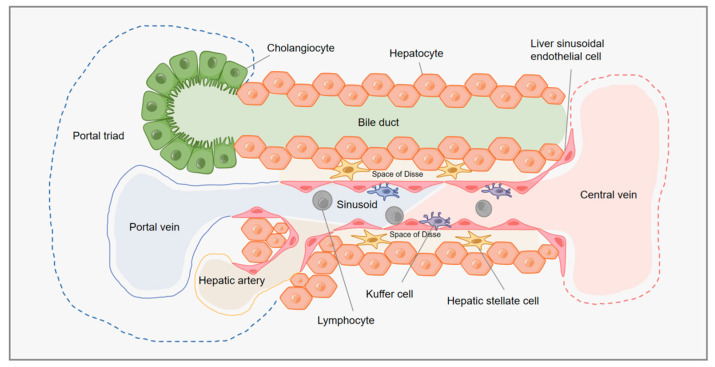
Schematic diagram of the liver structure. Hepatic sinusoids are encircled by liver sinusoidal endothelial cells (LSECs). Hepatocytes and endothelial cells are divided by the space of Disse, which also houses stellate cells. LSECs and hepatic macrophages (Kupffer cells) are in close proximity to one another. The interior of the bile duct tree is lined with cholangiocytes.

**Table 1 ijms-24-11592-t001:** The liver cell types and their primary functions.

Liver Cells	Types	Mass Percentage (%)	Functions	References
Parenchymal cells (PCs)	Hepatocytes and biliary tract cells	70–80	Managing the metabolism of nutrients and xenobiotics. Drug metabolism through phase I (CYP450), phase II, and phase III enzymes. Bile production and transport. Bilirubin conjugation and excretion. Production and recycling of proteins such as albumin. Glucose and lipid homeostasis. Production of hepatic clotting factors. Metabolism of amino acids. Cholesterol transport. Blood detoxification.	[81,82,83,84]
Nonparenchymal cells (NPCs)	Liver sinusoidal endothelial cells (LSECs)	15–20	Involved in the elimination of germs and endotoxins. Control over hepatic leukocyte migration. The primary cells that eliminate connective tissue molecules from the circulation.	[85,86,87]
	Kupffer cells (KCs)	15	Phagocytose cell and microbial debris. Phagocytose gut-derived microbial microorganisms that reach the liver through the bloodstream; Coordination of acute inflammation, host defense, and inflammation resolution. Maintenance of liver tissue balance and regeneration.	[21,85,88]
	Hepatic stellate cells (HSCs)	15	Metabolism of vitamin A. Storage of fat.	[89]

**Table 2 ijms-24-11592-t002:** Summary of the research on the applications of iPSCs in liver disease modeling.

Strategy	Abstract	Disease Model	Conclusions	References
2D differentiation strategy	Investigated whether iPSCs from α1-antitrypsin deficiency (ATD) individuals with or without severe liver disease could model these personalized variations in hepatic disease phenotypes.	Liver disease resulting from ATD	HLCs model the individual disease phenotypes of ATD patients with more rapid degradation of misfolded ATZ and lack of globular inclusions in cells from patients who have escaped liver disease.	[49]
	A library of human iPSCs lines were generated from individuals with a range of inherited metabolic disorders (IMDs), with a focus on 3 of the diseases, and hepatocytes were derived using human iPSCs from affected patients.	IMDs of the liver	Human iPSC-derived hepatocytes successfully recapitulate key features of the cytopathology seen in related diseases, such as aggregation of misfolded mutant α1-antitrypsin in the endoplasmic reticulum, deficient LDL receptor-mediated cholesterol uptake, and elevated accumulation of cellular lipids and glycogen.	[107]
3D differentiation strategy	Generated hepatic organoids that comprise different parenchymal liver cell types and have structural features of the liver using human pluripotent stem cells.	Nonalcoholic steatohepatitis (NASH)	Developed a hepatic organoid platform with human cells that can be used to model complex liver diseases, including NASH.	[40]
	Using 11 different healthy and diseased pluripotent stem cell lines, a reproducible method was developed to obtain multicellular human liver organs composed of hepatocytes, stellate cells, and Kupffer-like cells.	Steatohepatitis	Under free fatty acid treatment, the organoid reproduced the key features of steatohepatitis, including the steatosis, inflammation, and fibrosis phenotypes.	[77]

## Data Availability

Not applicable.

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
