# Peer review of "Pluripotent Stem Cell-Derived Hepatocyte-like Cells: Induction Methods and Applications"

_ijms, 2023, doi:10.3390/ijms241411592_

Round 1
Reviewer 1 Report
The manuscript by Qiulin Luo., et al, titled as“ Pluripotent stem cell-derived hepatocyte-like cells: induction methods and applications” is reviewed manuscript, authors have exntensively reviwed the published research on produceing hepatocyte like cells from pluripotent stem cells, focused on iPSCs. End-stage liver diseases need liver-transplantation, by applying this regenerative medicine approach, one can avoid the liver transplantation in future.
Here are my cooments to authors:
1. The manuscript in well written, has good amount of intraoduction and various methods to derive HLCs. I would suggest to add illustrtaed version of HLCs from stem cells and the growth factors, cell-signalling cascade involving in the HLCs production from stem cells (iPSCs/MSCs/ESCs)
2. Authors could consider adding a section on 3D bioprinting of liver, with the scope of HLCs derived from stem cells.
English is good, might require proof-read to check ny spelling/grammer mistakes
Author Response
Please see the PDF attachment.

Reviewer 2 Report
In their review Luo et al. focused on induced pluripotent stem cells (iPSCs) and their application in the field of hepatology.
Authors provide comprehensive review of state-of-the-art approaches regarding iPSCs differentiation into hepatocytes or organoid-like systems (more sophisticated 3D differentiation systems). Applications of iPSCs-derived hepatocyte-like cells/organoid are also discussed, focusing mainly on modeling of liver diseases and drug screening.
My feeling is that the review contains all important information, however, the main problem for me is the English language. Sentences are very difficult to understand, some words are missing, text flow is not smooth, etc.
When reading, I spent too much time trying to understand the sentences’ meaning and rather missed the overall included information.
I would recommend thorough, in-depth revision of whole text, focusing on not to repeat unnecessarily sentences and stress important information.
Minor points:
1) the last paragraph in the Introduction should be written in present tense.
2) Some sentences are repeatedly used many times – check carefully the text and omit all nonessential sentences
3) Double check references and their format: especially 20, 38, 49
see above
Author Response
Please see the PDF attachment.

Round 2
Reviewer 2 Report
The current version of the paper is significantly improved and authors addressed all my objections.
Some typos/suggestions remaining:
1) lines 79-82: in my opinion, the 2 sentences should be switched (if I understand correctly their meaning)
2) line 121: "is" should be deleted
3) lines 216-219: the sentence should be rephrased (suggestion: The combination of HGF and OSM factors with DEX increases the ability of hepatocytes to differentiate due to the fact that DEX upregulates the expression of mature liver-specific genes, such as the albumin.)
4) line 386: delete "of"
5) line 389: "improved" could be deleted
6) line 394: ... appears that THE induction ...
7) line 569: "making" should be deleted
8) line 587: I would delete also "In summary"
9) Table 1: I would correct the title (suggestion: "Liver cell types and their primary functions."). Separation between individual sections/items is not clear - better visual separation will improve the overall readability of the Table.
10) similarly to the Table 1, the separation of items in Table 2 should be still improved.
Author Response
Please see attached PDF with responses.
